# Evaluation of the Canonical Wnt Signaling Pathway in the Hearts of Hypertensive Rats of Various Etiologies

**DOI:** 10.3390/ijms25126428

**Published:** 2024-06-11

**Authors:** Maryla Anna Młynarczyk, Natalia Domian, Irena Kasacka

**Affiliations:** Department of Histology and Cytophysiology, Medical University of Bialystok, 15-222 Bialystok, Poland; maryla.mlynarczyk@sd.umb.edu.pl (M.A.M.); natalia.domian@umb.edu.pl (N.D.)

**Keywords:** Wnt/β-catenin, arterial hypertension, essential hypertension, secondary hypertension, myocardium

## Abstract

Wnt/β-catenin signaling dysregulation is associated with the pathogenesis of many human diseases, including hypertension and heart disease. The aim of this study was to immunohistochemically evaluate and compare the expression of the Fzd8, WNT1, GSK-3β, and β-catenin genes in the hearts of rats with spontaneous hypertension (SHRs) and deoxycorticosterone acetate (DOCA)-salt-induced hypertension. The myocardial expression of Fzd8, WNT1, GSK-3β, and β-catenin was detected by immunohistochemistry, and the gene expression was assessed with a real-time PCR method. In SHRs, the immunoreactivity of Fzd8, WNT1, GSK-3β, and β-catenin was attenuated in comparison to that in normotensive animals. In DOCA–salt-induced hypertension, the immunoreactivity of Fzd8, WNT1, GSK-3β, and β-catenin was enhanced. In SHRs, decreases in the expression of the genes encoding Fzd8, WNT1, GSK-3β, and β-catenin were observed compared to the control group. Increased expression of the genes encoding Fzd8, WNT1, GSK-3β, and β-catenin was demonstrated in the hearts of rats with DOCA–salt-induced hypertension. Wnt signaling may play an essential role in the pathogenesis of arterial hypertension and the accompanying heart damage. The obtained results may constitute the basis for further research aimed at better understanding the role of the Wnt/β-catenin pathway in the functioning of the heart.

## 1. Introduction

Arterial hypertension is a complex disease that affects a large part of the world’s population. In most cases, this disease develops slowly, over a long period of time, without noticeable symptoms. On the other hand, the complications of untreated hypertension can be life-threatening [1,2,3]. 

In 90% of cases, hypertension is primary and its cause is unknown. In 10%, arterial hypertension occurs secondary to other disorders of the body’s functioning [4,5]. The heart is the organ most affected by the complications of hypertension, which can cause impaired blood flow, unfavorable cardiomyocyte remodeling, and hypertrophy of the left ventricular wall, leading to heart failure [6,7]. An increase in blood pressure may be genetically determined or associated with environmental factors. 

In recent years, much attention has been paid to the Wnt/β-catenin signaling pathway, which is important for maintaining the homeostasis of the body [8,9]. The Wnt pathway is involved in many physiological as well as disease-related processes. The most important mediator of signal transduction in the canonical pathway is β-catenin. An increase in its concentration in the cytoplasm is associated with the activation of this pathway. In the absence of stimulation, the concentration of β-catenin is kept at a low level by its degradation in proteasomes [10,11]. Inhibition of the degradation complex involves adenomatous polyposis coli protein (APC), casein kinase 1 (CK1), and glycogen synthase kinase 3β (GSK-3β). Activating the Frizzled (Fzd) receptor leads to an increase in the level of β-catenin in the cytoplasm of the cell and its translocation to the nucleus, where it induces the expression of Wnt-dependent genes [10,12,13]. 

The Wnt/β-catenin signaling pathway participates in organogenesis and tissue homeostasis, and its dysregulation is associated with the pathogenesis of many human diseases, including hypertension and heart disease [12,14,15,16,17,18,19]. The studies conducted so far show that the Wnt/β-catenin signaling pathway plays a key role in the development of myocardial hypertrophy, myocardial fibrosis, ventricular remodeling, heart failure, and other pathophysiological processes [20,21].

Abnormalities in the functioning of organs directly responsible for blood pressure regulation may be related to dysregulated Wnt/β-catenin signaling [22,23]. 

Understanding the molecular mechanisms of structural and functional heart pathologies in hypertension and identifying physiological and exogenous factors that can modulate the course of this disease and influence a patient’s further prognosis seem particularly important for the development of modern and effective therapeutic strategies.

Considering the above, we decided to conduct research aimed at evaluating the main elements of the Wnt/β-catenin pathway in the hearts of hypertensive rats of various etiologies. The aim of this study was to immunohistochemically evaluate and compare the expression of the Fzd8, WNT1, GSK-3β, and β-catenin genes in the hearts of rats with spontaneous hypertension (SHRs) and DOCA–salt-induced hypertension.

## 2. Results

In rats in the study groups, systolic hypertension was found (SHRs: 160.8 ± 3.3 (WKY: 122.3 ± 2.3), DOCA–salt: 180.0 ± 13.0 (UNX: 126.0 ± 4.0)).

### 2.1. Immunohistochemistry

Positive immunohistochemical reactions for Fzd8, WNT1, GSK-3β, and β-catenin were observed in the hearts of all rats tested, although the severities of the reactions in the control and hypertensive rats were different.

The intensity of the immunohistochemical reaction showing Fzd8 was weaker in the hearts of the primary hypertensive rats (Figure 1B) and stronger in the secondary hypertensive rats (Figure 1D) compared to the control rats (Figure 1A,C).

The Wnt1 immunoreactivity in the hearts of the normotensive rats was strong (Figure 2A) or moderate (Figure 2C). The intensity of the reaction was attenuated in the hearts of the SHRs (Figure 2B) and enhanced in the DOCA–salt group (Figure 2D).

The hearts of the WKY rats showed stronger GSK-3β immunodetection (Figure 3A) compared to those observed in the SHR group (Figure 3B). The increase in secondary pressure (Figure 3D) led to an increase in GSK-3β immunoreactivity compared to the normotensive UNX group (Figure 3C).

In the hearts of all rats, the immunohistochemical reaction using an antibody against β-catenin was positive in intercalated discs (Figure 4). In the SHR hearts, β-catenin immunoreactivity was attenuated (Figure 4B) in relation to the WKY group (Figure 4A), while in the hearts of the secondary hypertensive rats (Figure 4D), this reaction was enhanced compared to the UNX group (Figure 4C). 

Morphometric image analysis results are presented in Table 1.

### 2.2. Real-Time PCR

In the hearts of the SHRs, decreases in the expression of the genes encoding Fzd8, WNT1, GSK-3β, and β-catenin were observed compared to the control group (WKY) (Figure 5), and the differences were statistically significant in the cases of WNT1 (Figure 5B) and β-catenin (Figure 5D). However, in the hearts of the rats with secondary hypertension, increases in the gene expression of all tested proteins were found in relation to the control group (UNX). The differences in the expression of the genes encoding Wnt1 (Figure 5B) and β-catenin (Figure 5D) were found to be statistically significant.

## 3. Discussion

Many factors and molecular pathways are involved in the pathogenesis of hypertension, the determination and explanation of which is an ongoing challenge for scientists and clinicians. Understanding the mechanisms of the multifactorial origin of hypertension and the various interactive regulations aimed at compensating for the action of vasoactive mediators is important in determining appropriate and effective treatments [1]. Untreated or ineffectively treated hypertension leads to damage to many organs. Important complications of hypertension include ischemic heart disease, cardiac arrhythmias, and heart failure, which pose serious risk of death [6]. 

In this study, we examined and compared the expression patterns of Fzd8, WNT1, GSK-3β, and β-catenin in the hearts of hypertensive rats of different etiologies. The results indicated differences in the expression of elements of the canonical Wnt pathway depending on the type of hypertension. In SHR hearts, the expression of all examined parameters was reduced (statistical significance was found for Wnt1 and β-catenin), while secondary hypertension caused increases in the expression of the investigated elements of Wnt/β-catenin signaling, which were especially significant for the Wnt1 ligand and β-catenin. 

In addition to the proven involvement of the canonical Wnt/β-catenin signaling pathway in the process of embryogenesis and the proliferation of cardiac muscle cells [15,16], it also plays an important role in the remodeling of the heart muscle and cardiomyocytes in pathological conditions of this organ [8,9,24]. In a study of the heart after ischemia–reperfusion injury, the involvement of the Wnt/β-catenin pathway in the processes of cardiac tissue remodeling in response to injury was demonstrated. Wnt1 was upregulated in the region of the injury and induced cardiac fibroblasts to proliferate and express pro-fibrotic genes, which prevented ventricular dilatation [25]. 

Mutual activation or inhibition of various pathways may be associated with the occurrence of hypertension. In hypertension, impaired activation of the renin–angiotensin–aldosterone system (RAAS), which is a critical regulator of blood volume, is often observed. Dysregulation of the Wnt signaling pathway may lead to disorders of the RAAS, consequently leading to disturbances in blood pressure homeostasis [26,27]. Research by Xiao et al. in an animal model showed an increase in the expression of Wnt pathway genes in rat kidneys in response to angiotensin II infusion. Moreover, the use of an inhibitor of the Wnt/β-catenin pathway reduced blood pressure [28]. According to the literature, in spontaneous hypertensive rats the plasma activity of the RAAS is increased and decreased in the course of secondary hypertension (DOCA–salt) [29]. In our study, we observed decreased expression in the Wnt/β-catenin pathway in the hearts of SHRs and observed intense activity in this pathway in DOCA–salt rats. Based on the obtained results, we expect additional mechanisms regulating the activity of the Wnt pathway in hypertensive rat hearts. 

The decrease in β-catenin expression in the SHR hearts presented in the current publication is consistent with reports by Zheng et al., who also found reduced β-catenin immunoreactivity in the intercalated discs of cardiomyocytes in an SHR group [30]. Other studies conducted using an experimental model of heart failure also showed decreases in β-catenin levels in the heart tissues of guinea pigs and hamsters [31,32]. 

Hypertrophy of cardiomyocytes is one of the main consequences of arterial hypertension. There is a large amount of evidence for the involvement of the Wnt/β-catenin pathway in cardiomyocyte remodeling processes. Research by Hirschy et al. showed that a persistent, stable peptide level, reflecting the activity of the Wnt/β-catenin pathway, caused the development of dilated cardiomyopathy and premature death in mice [33]. Another study demonstrated that genetic depletion of β-catenin significantly enhanced left ventricular function and survival in mice experiencing myocardial infarction, whereas stabilization of β-catenin had the opposite effect [34]. Xiang et al. proved that cardiac fibroblasts without β-catenin expression did not produce collagen when the Wnt/β-catenin pathway was activated under pressure-overload conditions. In a transgenic mouse model, this absence of β-catenin resulted in neither fibrosis nor cardiomyocyte hypertrophy [35]. Complications of primary hypertension in rat models, such as cardiomyocyte hypertrophy, may be observed due to the long-lasting state of elevated blood pressure, so the decreased activity of β-catenin could be considered an adaptive mechanism used to minimize hypertrophy. In this way, we could try to explain the decrease in the activity of the Wnt/β-catenin system in our study in the hearts of rats with essential hypertension. It is probable that additional signaling pathways inhibiting the Wnt/β-catenin cascade are also activated to provide protection against elevated blood pressure. There are reports indicating the role of Wnt/β-catenin pathway inhibitors in the prevention of cardiovascular diseases and their complications. The emerin protein found in skeletal muscle cells and cardiomyocytes is responsible, among other things, for regulating gene expression, maintaining the proper structure of the nucleus, and cell signaling. It has been shown that emerin maintains β-catenin at the proper level in the cell. A study on mice whose cardiomyocytes were deprived of emerin under conditions of pressure overload showed excessive expression of β-catenin and unfavorable tissue remodeling [36]. In another study, in the hearts of hypertensive rats subjected to angiotensin II (Ang II) infusions, high expression of β-catenin was observed. The above results suggest that the cardiomyocyte hypertrophy induced by Ang II is associated with excessive activity of the Wnt/β-catenin pathway. Simultaneously, the use of the β-catenin inhibitor ICG-001 blocks the expression of α-actin and myosin heavy chains (β-MHCs) in hypertrophic hearts [37].

Taking into account the limited number of reports indicating the relationship of the Wnt/β-catenin pathway with arterial hypertension, it is important and justified to investigate its involvement in this disease. The present study showed different changes in the Wnt/β-catenin signaling pathway in the hearts of rats with primary and secondary hypertension. This study was the first to evaluate components of the classical Wnt/β-catenin signaling pathway in various types of hypertension using immunohistochemical and molecular methods. In our work, we demonstrated that changes in the activity of the Wnt/β-catenin pathway in hypertension may occur at every level of this pathway. It should be noted that the intensity of the changes in the Wnt/β-catenin pathway in the heart depends on the etiology of the hypertension. The presented results may constitute the basis for further research aimed at better understanding the role of the Wnt/β-catenin pathway in the functioning of the heart, as well as the pathophysiological mechanisms leading to its dysfunction and complications in the state of elevated blood pressure. It is imperative to conduct additional research that may facilitate the integration of antihypertensive medications into clinical practice. Their mechanism of action would be based on inhibiting the Wnt pathway. A more detailed understanding of the role of the WNT/β-catenin pathway in hypertension-associated disturbances of heart functioning requires further investigation.

## 4. Materials and Methods

The research material came from the Department of Experimental Physiology and Pathophysiology, Medical University of Bialystok, courtesy of Professor Barbara Malinowska. The procedures performed on experimental animals were approved by the Local Ethics Committee for Animal Research in Olsztyn. Six-week-old male animals with initial weights of 170–200 g were kept at a constant humidity (60 ± 5%) and temperature (22 ± 1 °C). A 12/12-h light/dark cycle was maintained. The rats had free access to standard pelleted food and drinking water. 

The experimental animals were divided into four groups:SHRs—seven male rats with genetically determined systemic hypertension from an inbred strain established from Wistar rats selected for high blood pressure.WKY—five normotensive male Wistar Kyoto rats as a reference for the SHR group.DOCA–salt—seven male Wistar rats that underwent unilateral nephrectomies and were then rendered hypertensive using a salt-rich diet and Deoxycorticosterone Acetate (DOCA) injections.UNX—five normotensive male Wistar rats that were only uninephrectomized as a reference for the DOCA–salt-induced hypertensive rats.

### 4.1. DOCA–Salt Hypertension

The DOCA–salt animals were anesthetized by intraperitoneal injections of sodium pentobarbital (300 μmol or ~70 mg/kg body weight (bw)). Their right kidneys were removed through right lateral abdominal incisions. After a 1-week recovery period, hypertension was induced for 4 weeks by s.c. DOCA injections (67 μmol or ~25 mg/kg in 0.4 mL/kg dimethylformamide (DMF)) twice weekly and replacing their drinking water with a 1% sodium chloride (NaCl) solution. The normotensive control rats that received unilateral nephrectomies (UNX) also had their kidneys removed but received subcutaneous DMF (0.4 mL/kg) twice a week and drank tap water.

### 4.2. Indirect Blood Pressure Measurement

After 6 weeks, the systolic blood pressure was measured in all animals by a non-invasive tail cuff method (rat tail blood pressure monitor, Hugo Sachs Elektronik-Harvard Apparatus, March-Hugstetten, Germany). The measurements were considered reliable if three consecutive measurements did not differ by more than 5 mmHg. Then, the average was taken. Hypertension (systolic blood pressure (SBP) values equal to or greater than 150 mmHg) was verified in the SHRs and DOCA–salt animals.

### 4.3. Collection and Fixation of Material

After 6 weeks, heart muscle fragments were collected from all rats under deep anesthesia with pentobarbital (50 mg/kg body weight). Cardiac tissue was immediately fixed in 4% buffered formalin and routinely embedded in paraffin or placed in an RNAlater solution (AM7024 Thermo Fisher, Waltham, MA, USA) and stored at −80 °C. Paraffin blocks were cut into 4 μm sections and stained with hematoxylin and eosin for general histological evaluation. Immunohistochemical reactions were performed to detect Fzd8, Wnt1, GSK-3β, and β-catenin. The samples stored in the RNAlater solution were analyzed by real-time PCR to assess the expression of the genes encoding Fzd8, Wnt1, GSK-3β, and β-catenin.

### 4.4. Immunohistochemistry

Immunostaining was carried out by the following protocol (Kasacka et al., 2018) [38]: Paraffin-embedded sections were deparaffinized and hydrated in pure alcohols. Sections of the left ventricles of the hearts were subjected to pretreatment in a pressure chamber and heated using appropriate Target Retrieval Solutions (Citrate (pH = 6.0) (S 2369; Agilent Technologies, Inc., Santa Clara, CA, USA) (for β-catenin) and TRS (pH = 9.0) (S 2367; Agilent Technologies, Inc., Santa Clara, CA, USA) (for Fzd8, Wnt1, and GSK-3β)). After cooling to room temperature, the sections were incubated with Dako REAL Peroxidase-Blocking Solution (S 2023; Agilent Technologies, Inc. Santa Clara, CA, USA). The sections were incubated with the primary antibodies for Fzd8 (1:400; ab155093, Abcam, Cambridge, UK), Wnt-1 (1:500; ab189001, Abcam, Cambridge, UK), GSK-3β (1:100; ab68476, Abcam, Cambridge, UK), and β-catenin (1:2000; ab32572, Abcam, Cambridge, UK) for 24 h at +4 °C in a humidified chamber. This procedure was followed by incubation with the secondary antibody (REAL™ EnVision™ Detection System, Peroxidase/DAB, Rabbit/Mouse detection kit (K5007; Agilent Technologies Denmark Ap/S, Produktionsvej 42, 2600 Glostrup, Denmark)). The bound antibodies were visualized by incubation with DAB Flex chromogen. Finally, the left ventricles of the heart sections were counterstained with hematoxylin QS (H-3404 Vector Laboratories, Burlingame, CA, USA) and observed under a light microscope. The sections were dehydrated, and the specificity of the antibodies was confirmed using a negative control, where the antibodies were replaced by Antibody Diluent (S3022; Agilent Technologies Denmark Ap/S, Produktionsvej 42, 2600 Glostrup, Denmark). The staining results were evaluated and documented using an Olympus BX43 light microscope (Olympus 114 Corp., Tokyo, Japan) with an Olympus DP12 digital camera (Olympus 114 Corp., Tokyo, Japan).

### 4.5. Real-Time PCR

Tissue fragments taken from the left ventricles of the rats’ hearts were placed in an RNA-later solution. Total RNA was isolated using a NucleoSpin^®^ RNA Isolation Kit (Machery-Nagel, Oensingen, Switzerland). The quantification and quality control of the total RNA were carried out using a NanoDrop 2000 spectrophotometer (ThermoScientific, Waltham, MA, USA). A 1 µg aliquot of total RNA was reverse-transcribed into cDNA using an iScript™ Advanced cDNA Synthesis Kit for RT-qPCR (BIO-RAD, Barkley, CA, USA). The synthesis of the cDNA was performed in a final volume of 20 μL using a Thermal Cycler (SureCycler 8800, Aligent Technologies, Santa Clara, CA, USA). For reverse transcription, the mixtures were incubated at 46 °C for 20 min, heated to 95 °C for 1 min, and finally cooled quickly at 4 °C. Quantitative real-time PCR reactions were performed using Stratagene Mx3005P (Aligent Technologies) with the SsoAdvanced™ Universal SYBER^®^ Green Supermix (BIO-RAD). Specific primers for FZD8 (*FZD8*), Wnt1 (*WNT1*), Gsk3β (*GSK3β*), Ctnnb1 (*CTNNB1*), and GAPDH (*GAPDH*) were designed by BIO-RAD. The housekeeping gene GAPDH (*GAPDH*) was used as a reference gene for quantification. To determine the expression levels of the test genes, standard curves were constructed separately for each gene with serially diluted PCR products. The PCR products were obtained by cDNA amplification using the following specific primers: *FZD8* (qRnoCED0054913, BIO-RAD), *WNT1* (qRnoCED0003949, BIO-RAD), *GSK3β* (qRnoCID0001683, BIO-RAD), *CTNNB1* (qRnoCID0053256, BIO-RAD), and *GAPDH* (qRnoCID0057018, BIO-RAD). qRT-PCR was carried out in a doublet in a final volume of 20 μL under the following conditions: 2 min of polymerase activation at 95 °C, 5 s of denaturation at 95 °C, and 30 s of annealing at 60 °C for 35 cycles. The PCR reactions were checked, including no-RT controls, the omission of templates, and melting curves, to ensure that only one product was amplified. The relative quantification of gene expression was carried out by comparing the Ct values using the ΔΔCt method. All results were normalized to *GAPDH*.

### 4.6. Measurement of the Intensity of the Immunohistochemical Reaction

Sections of the heart were taken from each animal for immunohistochemistry showing Fzd8, Wnt1, GSK-3β, and β-catenin. Five randomly selected microscopic fields (each field measured 0.785 mm^2^ at 200× magnification (20× lens and 10× eyepiece)) from each heart section were documented using an Olympus DP12 microscope camera. Nikon’s NIS Elements Advanced Research microscopy image analysis software (version 3.10) was used to evaluate digital images of the heart samples. The average optical densities of the examined objects were measured to assess the intensities of the immunohistochemical reactions. The intensities of the immunohistochemical reactions for Fzd8, Wnt1, GSK-3β, and β-catenin were measured in each image and quantified using a grayscale level of 0–256. A value of 0 meant a completely white pixel, i.e., minimum light saturation, while 256 meant a completely black pixel, i.e., maximum light saturation.

### 4.7. Statistical Analysis

All data collected for the individual rats were assigned to two control groups (WKY and UNX) and two treatment groups (SHRs and DOCA–salt). For measurable features, the arithmetic means and standard errors (SEs) were calculated. Then, using the STATISTICA 13.3 computer package, statistical analysis was performed using a one-way ANOVA test. Fisher’s Least Significant Differences test was used to perform post hoc analysis. The level of statistical significance was assumed to be *p* < 0.05.

## Figures and Tables

**Figure 1 ijms-25-06428-f001:**
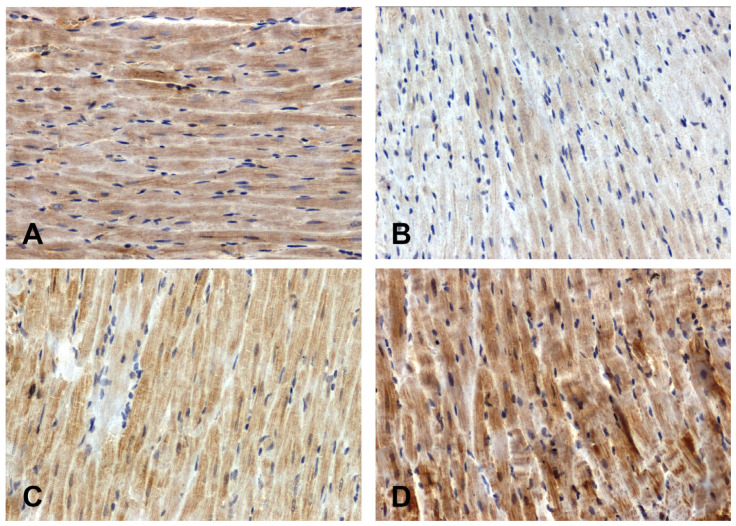
Immunohistochemical analysis of Fzd8 in the hearts of the rats: (**A**)—WKY, (**B**)—SHR, (**C**)—UNX, (**D**)—DOCA–salt.

**Figure 2 ijms-25-06428-f002:**
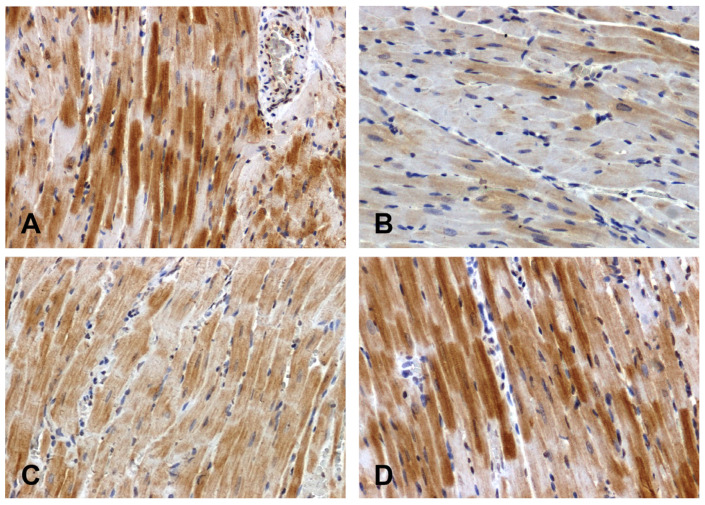
Immunohistochemical analysis of Wnt1 in the hearts of the rats: (**A**)—WKY, (**B**)—SHR, (**C**)—UNX, (**D**)—DOCA–salt.

**Figure 3 ijms-25-06428-f003:**
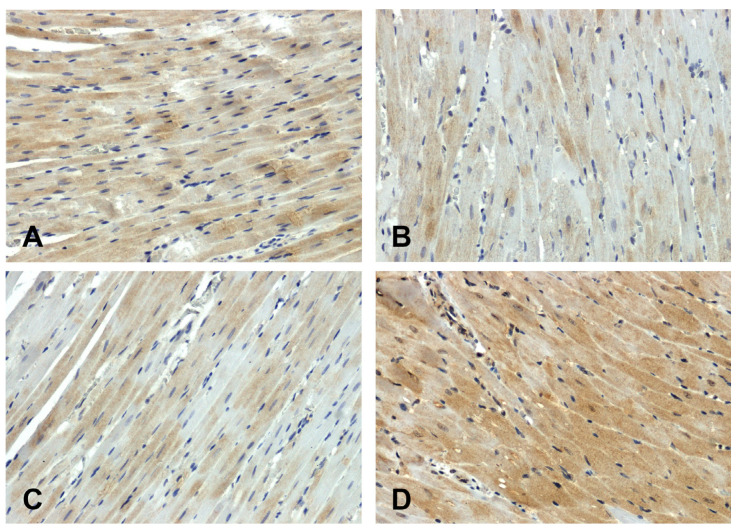
Immunohistochemical analysis of GSK-3β in the hearts of the rats: (**A**)—WKY, (**B**)—SHR, (**C**)—UNX, (**D**)—DOCA–salt.

**Figure 4 ijms-25-06428-f004:**
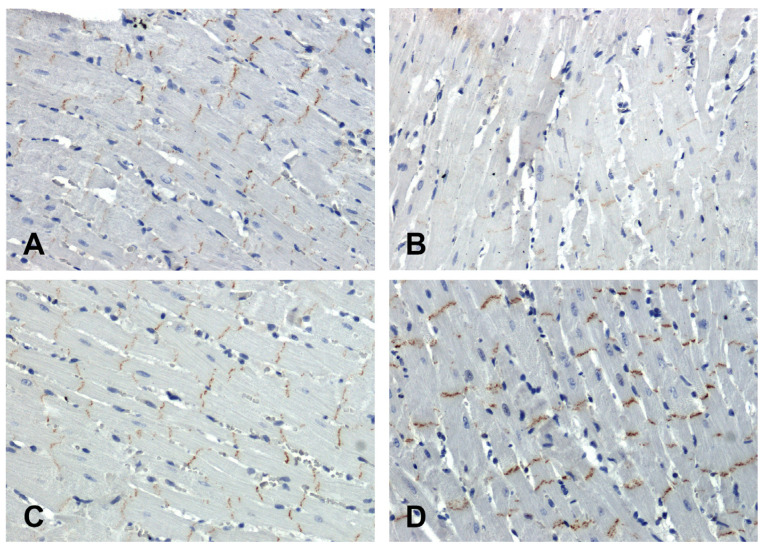
Immunohistochemical analysis of β-catenin in the hearts of the rats: (**A**)—WKY, (**B**)—SHR, (**C**)—UNX, (**D**)—DOCA–salt.

**Figure 5 ijms-25-06428-f005:**
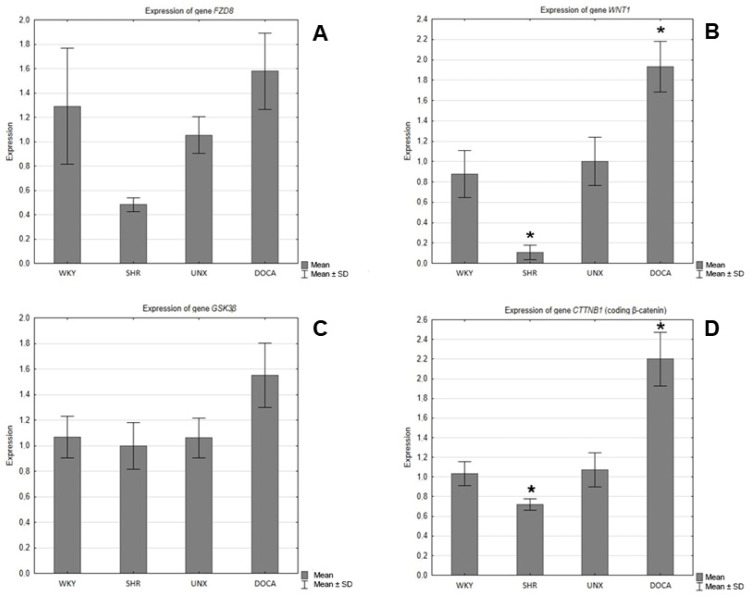
Expression of genes encoding (**A**) FZD8, (**B**) WNT1, (**C**) GSK-3β, and (**D**) β-catenin in hearts of normotensive and hypertensive rats (* *p* < 0.05).

**Table 1 ijms-25-06428-t001:** Intensity of immunoreactions determining Fzd8, WNT1, GSK-3β, and β-catenin in hearts of normotensive and hypertensive rats (means ± SEs).

Groups of Rats	Intensity of Immunohistochemical Reaction in Heart Scale from 0 (White Pixel) to 255 (Black Pixel)			
	Fzd8	WNT1	GSK-3β	β-Catenin
WKY	143.2 ± 4.12	157.72 ± 5.99	93.67 ± 1.87	157.27 ± 1.98
SHRs	64.52 ± 8.37 *↓	72.79 ± 7.35 *↓	67.82 ± 2.97 ↓	86.29 ± 4.63 *↓
UNX	110.67 ± 4.51	116.84 ± 2.61	75.08 ± 5.23	162.88 ± 1.75
DOCA–salt	174.67 ± 5.58 *↑	168.09 ± 6.97 *↑	136.65 ± 6.45 *↑	183.67 ± 1.57 *↑

* *p* < 0.05 (hypertension vs. control); **↑**—intensification of immunohistochemical reaction; **↓**—weakening of immunohistochemical reaction.

## Data Availability

The data presented in this study are available on request from the corresponding author.

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
