# Peer review of "Evaluation of the Canonical Wnt Signaling Pathway in the Hearts of Hypertensive Rats of Various Etiologies"

_ijms, 2024, doi:10.3390/ijms25126428_

Round 1
Reviewer 1 Report
Comments and Suggestions for Authors
The manuscript deals with Wnt signaling pathway in the heart of rats with primary and secondary hypertension. The manuscript may contribute to further examination of the Wnt/beta catenin pathway in the heart functioning and pathology; however, there are certain issues which need to be addressed and clarified.
Introduction is comprehensive and brief, stating the relevant facts for the investigated subject.
The applied methods and experimental design are standard for this type of investigation and they are described in detail.
Results: The results in the Section 2.2 regarding RT PCR are double presented, as Fig. 5 and Table 2. Besides, the values in the table does not correspond to the values for WTN1, which raises the question which values is correct.
Discussion is comprehensively written, with enough data from other studies to corroborate the observed findings.
Comments on the Quality of English LanguageEnglish language needs minor editing.
Author Response
I would like to thank Reviewer 1 for the time devoted to analyzing and assessing our work and for his valuable comments. In response to further questions, I would like to inform you that:
Ad.1. Results: The results in the Section 2.2 regarding RT PCR are double presented, as
Fig. 5 and Table 2. Besides, the values in the table does not correspond to the values
for WTN1, which raises the question which values is correct
I sincerely apologize for the error that occurred while transcribing the data and, of course, as
suggested by the Reviewer, Table 2 was deleted.
Reviewer 2 Report
Comments and Suggestions for Authors
This manuscript by MÅ‚ynarczyk et al. describes the evaluation of canonical Wnt signaling pathway in hearts of hypertensive rats. Wnt/β-catenin signaling dysregulation is important to many human diseases. The authors used immunohistochemistry and qPCR to compare the protein and mRNA levels of Fzd8, WNT1, GSK-3β and β-catenin in the hearts of rats with SHR and DOCA-salt-induced hypertension. Despite that the article contains important observations, more in-depth experiments and analysis are needed. For example, is there any potential explanation for the difference between gene expression level and the protein level? The authors mentioned that other additional mechanisms could also regulate the activity of Wnt signaling pathway. What are those additional mechanisms? What’s the potential application of this study? There are a few studies using β-catenin inhibitors to prevent hypertensive heart disease. This could be a good aspect to discuss.
Minor point: The citation format needs to be consistent.
Author Response
According to Referee 2 suggestions I would like to inform that:
Ad.1. Is there any potential explanation for the difference between gene expression level and the protein level?
Discrepancies between gene expression levels assessed by real-time PCR (qPCR) and protein levels detected by immunohistochemistry (IHC) can be attributed to several biological factors:
- Post-transcriptional regulation: mRNA levels determined by qPCR do not always directly correlate with protein levels due to post-transcriptional mechanisms such as mRNA stability, translational efficiency, and protein degradation. Regulatory processes such as splicing, editing, and the activity of noncoding RNAs (e.g., microRNAs) can significantly alter the rate of translation of mRNAs into functional proteins.
- Protein turnover: Protein stability and degradation rates, regulated by the ubiquitin-proteasome system or lysosomal pathways, influence protein abundance regardless of mRNA level. Thus, even if mRNA levels are high, rapid protein turnover may result in low protein expression detectable by IHC.
- Temporal and spatial expression: qPCR provides an average measure of mRNA levels in a bulk tissue sample, while IHC can reveal spatial heterogeneity and localized protein expression in specific cell types or tissue regions. Differences in temporal mRNA and protein expression patterns resulting from different cellular responses and signaling pathways may further contribute to the discrepancies.
- Quantitative sensitivity and dynamic range: qPCR is highly sensitive and can detect low abundance mRNAs, often with a wide dynamic range. In contrast, IHC is semi-quantitative and may not effectively detect low-abundance proteins. The visual scoring or image analysis methods used in IHC are less precise compared to the quantitative measurements provided by qPCR.
- Sample heterogeneity: Differences in cell composition in tissue samples may affect the correlation between mRNA and protein levels. For example, immune cell infiltration or stromal content may differentially express a gene of interest, influencing qPCR results, while IHC may selectively highlight expression in specific cell types.
Ad.2. The authors mentioned that other additional mechanisms could also regulate the activity of Wnt signaling pathway. What are those additional mechanisms?
Other mechanisms influencing the activity of the Wnt signaling pathway:
- post-translational modifications (also lipid) of Wnt ligands. All Wnt proteins undergo N-glycosylation, which is necessary for their proper secretion to the cell membrane surface membrane availability of Fzd receptors, which may be influenced by other proteins (R-spondin and Norrin protein) or ubiquitin ligases (RNF43 and ZNRF3).
- overexpression of cadherins may lead to inhibition of the canonical Wnt pathway by relocalization of β-catenin to the cell membrane
- activity of the so-called complex marking β-catenin for degradation (various LRP6 domain phosphorylation enzymes) GSK-3β
The canonical Wnt signaling pathway can be activated by many mechanisms related to ligand binding and the expression of internal and external regulatory elements, but the common and key component is the protein β-catenin, the accumulation of which in the cytoplasm results in translocation to the cell nucleus and transcription of numerous target genes.
Ad.3. What’s the potential application of this study? There are a few studies using β-catenin inhibitors to prevent hypertensive heart disease. This could be a good aspect to discuss.
Knowledge of the functional structure of the Wnt signaling pathway allows us to search for therapeutic targets to inhibit the transcriptional activity of β-catenin.
In order to use changes in the activity of the canonical Wnt pathway therapeutically, it is necessary to have detailed knowledge of its structure and the principles of signal transduction. This, in turn, may allow the search for effective therapeutic targets aimed at reducing the transcriptional activity of β-catenin, which may potentially increase the therapeutic effect.
Therefore, understanding the biological role of the Wnt/β-catenin pathway in hypertension may be crucial in the treatment and diagnosis of heart disease to improve the quality of life of patients with heart disease.
As suggested by the Reviewer, the final part of the Discussion includes information on certain inhibitor proteins of the canonical Wnt signaling pathway.
Minor point: The citation format needs to be consistent
I kindly inform you that the manuscript has been checked by native English and corrected and literature items have been unified.
Because the numbering of the lines in the manuscript has changed, the corrections made are marked in color in the text
Yours sincerely
Irena Kasacka
Round 2
Reviewer 1 Report
Comments and Suggestions for Authors
The manuscript is corrected according to suggestions.
Reviewer 2 Report
Comments and Suggestions for Authors
Thanks for addressing my questions.